# Unsupervised Point Cloud Completion and Segmentation by Generative Adversarial Autoencoding Network

**Changfeng Ma**
Nanjing University
changfengma@smail.nju.edu.cn

**Yang Yang**
Nanjing University
yyang_nju@outlook.com

**Jie Guo**
Nanjing University
guojie@nju.edu.cn

**Fei Pan**
Nanjing University
panfei@smail.nju.edu.cn

**Chongjun Wang**
Nanjing University
chjwang@nju.edu.cn

**Yanwen Guo**[*]
Nanjing University
ywguo@nju.edu.cn

## Abstract

Most existing point cloud completion methods assume the input partial point cloud is clean, which is not the case in practice, and are generally based on supervised learning. In this paper, we present an unsupervised generative adversarial autoencoding network, named UGAAN, which completes the partial point cloud contaminated by surroundings from real scenes and cutouts the object simultaneously, only using artificial CAD models as assistance. The generator of UGAAN learns to predict the complete point clouds on real data from both the discriminator and the autoencoding process of artificial data. The latent codes from generator are also fed to discriminator which makes encoder only extract object features rather than noises. We also devise a refiner for generating better complete cloud with a segmentation module to separate the object from background. We train our UGAAN with one real scene dataset and evaluate it with the other two. Extensive experiments and visualization demonstrate our superiority, generalization and robustness. Comparisons against the previous method show that our method achieves the state-of-the-art performance on unsupervised point cloud completion and segmentation on real data.

## 1 Introduction

In recent years, point clouds have gained more popularity [11] as the standard outputs of 3D scanning devices [17, 39] and the fundamental data structure to represent and process 3D data [7, 6, 19]. However, clean and complete point clouds of objects, on which the downstream applications such as reconstruction [13] significantly rely, are hard to obtain in practice due to the nature of scanners and occlusions. Point cloud completion which infers a complete object model given a partial point cloud thus has received considerable attention. Recent supervised methods [42, 41, 31, 26, 36, 34] show remarkable performance on recovering the original shapes from incomplete point clouds, but generally assume that the input point cloud is clean and does not contain any noises and outliers. Such an assumption does not always hold in practice, because the surroundings around the target object will be inevitably seen by the cameras during scanning real scenes. Though semantic or instance segmentation on point clouds has been investigated in the literature, the state-of-the-art instance segmentation methods [21, 5] only report around 68% average precision on the public ScanNet [7] and  [1] datasets. That is to say, accurately cutting out the target object from the whole point cloud

---

[*]Corresponding author.

36th Conference on Neural Information Processing Systems (NeurIPS 2022).

still remains challenging. Therefore, it is important for the downstream applications to remove noises from point clouds or directly complete the point clouds with noises.

In this paper, we propose an unsupervised end-to-end network for completing the partial point cloud with noises from real scenes and outputting the object mask at the same time, dubbed UGAAN. Previous supervised point cloud completion and segmentation works require the paired clean and complete point clouds, together with object-level point labels of real scenes, for training, which are hard and costly to obtain in practice [39, 22]. Most deep-learning-based methods heavily rely on such training data, which limits their applications in practice. Clearly different from previous methods, our UGAAN only needs unlabeled real-scene objects and unpaired artificial CAD models for unsupervised training.

We employ the framework of GAN [10] which is widely used in unsupervised tasks. Traditional framework of GAN is hard to converge due to the gap between real-scene data and artificial data. To resolve this problem, we utilize an autoencoding generator to learn the basic shapes of objects from the artificial data for easier convergence. Usually, the discriminator could easily distinguish the artificial data as real data and the prediction as fake data, leading to bad performance due to the data gap between predicted point clouds and artificial data. Our discriminator, by contrast, accepts the prediction of artificial data rather than the artificial data itself as real data, which can reduce the negative effect of data gap. Our discriminator also takes latent codes extracted from real-scene data and artificial data as the criterion, by which the generator can robustly learn a better latent space for prediction. We employ a refiner to refine and upsample the first prediction result, and the final complete point clouds thus generated are more accurate and uniform. Finally, the segmentation module predicts the labels according to the predicted point cloud and input point cloud. Compared with usual point cloud completion datasets [42] whose objects are aligned by the yaw angles and scales, real-scene datasets such as ScanNet and S3DIS [1] contain objects that have different yaw angles and varied scales, making the prediction of complete point clouds challenging. To accommodate real-scene data, we randomly rotate and scale the artificial point clouds for easier learning and convergence.

We train our UGAAN on ScanNet with ShapeNet as the artificial dataset without any pre-training or fine-tuning, and then directly evaluate it on the other two real-scene datasets, including S3DIS and ScanObjectNN [28]. Comparisons with the state-of-the-art methods [22, 39] show the superiority of our method.

The main contributions of our work are as follows.

- We propose, for the first time, an unsupervised end-to-end network for completion and segmentation of point clouds with outliers by combining GAN and an autoencoding network.

- Our method completes the real-scene object with outliers from the background without the need of any pre-training.

- We conduct extensive experiments for analysis and comparisons on three different real-scene datasets, the results of which show the superiority and generalization ability of our methods.

The rest of this paper is organized as follows. Section 2 reviews briefly the works on point cloud completion and segmentation. Section 3 introduces our UGAAN in detail. Dataset generation, experiments and comparisons are shown in Section 4 and Section 5. Section 6 concludes the whole paper and highlights future work.

## 2 Related Work

### 2.1 Point cloud completion and segmentation.

In the early stage, existing successful works [27, 8] utilize voxels as representations of 3D models on which 3D convolutional neural networks can be immediately applied. PMP-Net [34] and PMP-Net++ [33] generate complete point clouds by moving input points to appropriate positions iteratively with minimum moving distance. SnowflakeNet [36] utilizes transformer and point-wise feature deconvolutional modules to refine the first-stage point could multiple times. For semantic segmentation, Jiang *et al.* [15] propose a multi-stage ordered convolution module to stack and encode the information from eight spaces, achieving orientation encoding and scale awareness. Liang *et*

*al.* [21] split nodes of a pre-trained, intermediate, semantic superpoint tree for proposals of instance objects. Point Transformer [43] introduces transformer in the encoding process of point cloud to learn the representation, which can be applied to different point cloud processing such as segmentation and classification.

## 2.2 Unsupervised point cloud learning.

Xiao *et al.* [37] summarize the unsupervised point cloud representation learning works using DNNs. Some works [20, 38, 30] propose point cloud unsupervised pre-training approaches, Others proposed unsupervised model [40, 12, 4] achieves excellent results on classification, segmentation and upsampling tasks. Jiang *et al.* [14] introduce two contrastive losses to respectively facilitate downstream classification and segmentation. Coseg [22] presents a point cloud object co-segmentation task, aiming to segment the common 3D objects in a set of point clouds, and fathom a method with co-contrastive losses to minimize feature discrepancy inside estimated object points and maximize feature separation between the object and background points, which also leads to a weak generalization. Chen *et al.* [39] propose an unpaired point cloud completion method that can be trained without requiring explicit correspondence between partial and complete point clouds by employing an adaptive transform network in GAN, however, the training process of its network is complicated with several pre-training models.

# 3 Method

## 3.1 Problem Formulation

Let $P_r \in \mathbb{M}_{n_0 \times 3}$ represent the point cloud from real-scene data with distribution $p_r$ and $P_a$ denote artificial data with distribution $p_a$, respectively. $P_r$ contains points belonging to the object and from background (noises). $Obj(P_r)$ represents the object points on $P_r$. Assuming that $Obj(P_r) \sim p_o$, the goal of unsupervised point cloud completion and segmentation is to train a generator $G$ satisfying the following two conditions:

1. $G(P_r; \theta_g) \sim p_a$,
2. $G(P_r; \theta_g) \cap P_r \sim p_o$,

where $G(P_r; \theta_g)$ represents the point cloud predicted by $G$ with input $P_r$ and parameters $\theta_g$, and the operation $\cap$ denotes the intersection of two point clouds[2]. The two conditions are utilized to constrain completion and segmentation processes of point clouds, respectively. Condition 1 encourages the predicted shapes to be similar to the shapes in the artificial dataset. Condition 2 requires that the generator should predict shapes according to $P_r$ rather than generating random shapes, and the segmentation result which is the intersection of prediction and $P_r$ should match the object points.

## 3.2 Network Architecture

We employ Generative Adversarial Network (GAN) [10] to solve this unsupervised learning problem. As shown in Figure 1, our UGAAN consists of an autoencoding generator and a discriminator that plays a minimax game for unsupervised completion and segmentation of real-scene object point cloud. Furthermore, we also utilize a refiner to refine the predicted complete point cloud and a segmentation module for label prediction.

**Autoencoding Generator.** We use PointNet++ [24] as the encoder $E$ of the generator to extract the latent code $\mathbf{z}$ of input point cloud $P_i$, and a multi-layer perceptron (MLP) as the decoder to predict the point cloud $P_o \in \mathbb{M}_{n_1 \times 3}$. The generator takes artificial data as input to learn basic shapes of point clouds. Then, we employ a lightweight encoder-decoder refiner [42] $R$ on $P_o$ to get the refined and upsampled output $P_{o^r} \in \mathbb{M}_{n_2 \times 3}$. Different from [42], the upsampling module directly refines the complete point cloud $P_o$. Given an input point cloud $P_i$, we can obtain the latent code, the complete prediction and the refinement by $\mathbf{z} = E(P_i; \theta_g)$, $P_o = G(P_i; \theta_g)$ and $P_{o^r} = R(P_o; \theta_r)$, respectively, where $\theta_g$ and $\theta_r$ are the parameters of generate $G$ and refiner $R$.

---

[2]A point cloud $P \in \mathbb{M}_{n \times 3}$ can also be seen as a set containing $n$ points $\mathbf{p} \in \mathbb{M}_{1 \times 3}$.

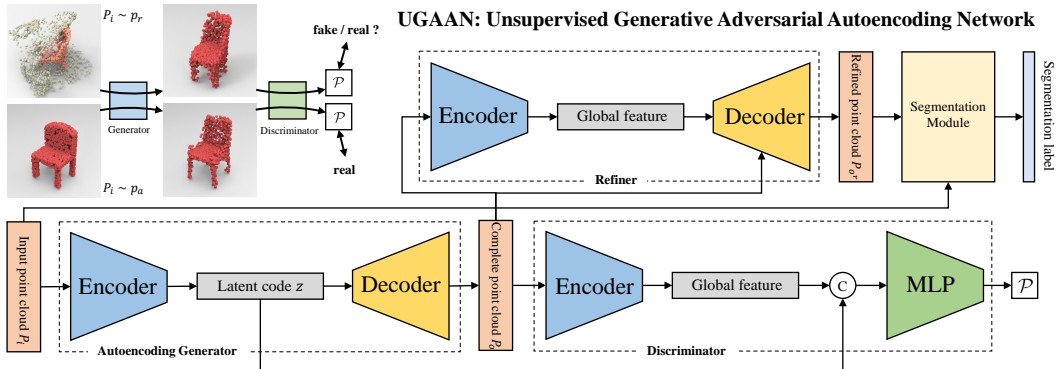

Figure 1: The overall architecture of our UGAAN. The network contains an autoencoding generator, a discriminator, a refiner and a segmentation module. The autoencoding generator predicts the complete point cloud according to the input. The discriminator takes predicted point cloud and latent code as the criterion and decides whether the input point cloud is from artificial data or not. The refiner refines the complete point cloud, and the segmentation module predicts labels.

**Discriminator.** We take advantage of the encoder proposed in [42] to extract the global features $\mathbf{f}$ of $P_o$. Then the global feature $\mathbf{f}$ and the latent code $\mathbf{z}$ are combined and fed into a MLP for predicting the probability $\mathcal{P}$:

$$\mathcal{P}(P_i; \theta_g, \theta_d) = D(\mathbf{z}, P_o; \theta_d) = D\big(E(P_i; \theta_g), G(P_i; \theta_g); \theta_d\big), \tag{1}$$

where $P_i \sim p_a$ and $\theta_d$ is the parameters of the discriminator. To reduce the negative effect of the shape gap between the prediction and artificial data, we feed the predicted shapes rather than the artificial data into the discriminator. The discriminator also takes latent codes as the criterion by which the generator can learn better latent codes for prediction.

**Segmentation Module.** Finally, the segmentation results (i.e. the object points of $p_r$) are obtained under the guidance of the complete prediction $P_o$. Here, we directly calculate the intersection between $P_o$ and $P_r$ to get the object points, rather than predicting with another network. Due to the discrete nature of point clouds, the strict intersection between $P_o$ and $P_r$ is almost empty. As a consequence, we make use of general intersection between the point clouds $P_1$ and $P_2$ which is indicated by the overlapping area:

$$P_1 \cap_d P_2 = \{cd_{\mathbf{p} \to P_1} < d \wedge cd_{\mathbf{p} \to P_2} < d | \mathbf{p} \in \mathbb{M}_{1 \times 3}\}, \tag{2}$$

where $cd_{\mathbf{p} \to P_1}$ will be introduced in Equation 5 with $d$ being the threshold. The predicted segmentation label of $\mathbf{p} \in P_r$ conditioned on $P_o$ with threshold $d$ can be computed as:

$$S_d(\mathbf{p}; P_r, P_O) = \begin{cases} 1 & \mathbf{p} \in P_r \cap_d P_o, \\ 0 & others, \end{cases} \tag{3}$$

Besides, the segmentation label of $P_r = [\mathbf{p}_1^T, ..., \mathbf{p}_j^T, ..., \mathbf{p}_n^T]^T$ can be obtained by $S_d(P_r; P_o) = [S_d(\mathbf{p_1}; P_r, P_O), ..., S_d(\mathbf{p_j}; P_r, P_O), ..., S_d(\mathbf{p_n}; P_r, P_O)]^T \in \mathbb{M}_{n \times 1}$.

### 3.3 Optimization

We modify the optimization strategy introduced by GAN [10] to train our UGAAN on the unsupervised point cloud completion and segmentation problem. Specifically, the optimization problem of the proposed UGAAN is formulated as:

$$\min_{\theta_g} \max_{\theta_d} \mathop{\mathbb{E}}_{P_i \sim p_a} \left[ \log \mathcal{P}(P_i; \theta_g, \theta_d) \right] + \mathop{\mathbb{E}}_{P_i \sim p_r} \left[ \log \left( 1 - \mathcal{P}(P_i; \theta_g, \theta_d) \right) \right]. \tag{4}$$

Due to the data gap between artificial data and real-scene data [39] and the limited fitting ability of the decoder, the terms $G(P_r)$ and $P_a$ often exhibit huge differences as shown in the top left part of Figure1. It makes $D$ easier to optimize than $G$ which makes network hard to converge and will result in poor completion performance. Thus, different from the original GAN where the

discriminator directly works on $G(P_r)$ and $P_a$, the discriminator of UGAAN predicts according to $G(P_r)$ and $G(P_a)$ to reduce the negative effects caused by these gaps for easier convergence. Besides, if we impose no other constraints on problem 4, the generator may just generate random points after the optimization process is complete. Thus, we also impose two losses on problem 4 to restrict the prediction of the generator, namely the autoencoding loss between $G(P_a)$ and $P_a$, and the reconstruction loss between $G(P_r)$ and $P_r$.

We employ the well-known Chamfer Distance(CD) [9] to measure the similarity between point clouds. For a point cloud $P = [\mathbf{p}_1^T, ..., \mathbf{p}_j^T, ..., \mathbf{p}_n^T]^T$, where $\mathbf{p}_j \in \mathbb{M}_{1 \times 3}$, and a point $\mathbf{p} \in \mathbf{M}_{1 \times 3}$, the Chamfer Distance from $\mathbf{p}$ to $P$ is:

$$cd_{\mathbf{p} \to P} = \min_{j=1}^{n} ||\mathbf{p} - \mathbf{p}_j||^2. \tag{5}$$

For two point clouds $P_1 = [\mathbf{p}_1^T, ..., \mathbf{p}_j^T, ..., \mathbf{p}_n^T]^T$ and $P_2$, the Chamfer Distance from $P_1$ to $P_2$ is:

$$cd_{P_1 \to P_2} = \frac{1}{n_1} \sum_{j=1}^{n} cd_{\mathbf{p}_j \to P_2}. \tag{6}$$

Besides, the Chamfer Distance between $P_1$ and $P_2$ is defined as:

$$cd_{P_1 \leftrightarrow P_2} = cd_{P_1 \to P_2} + cd_{P_2 \to P_1}. \tag{7}$$

The point cloud $P_1$ will be a part of the point cloud $P_2$ when the term $cd_{P_1 \to P_2}$ is minimized. Moreover, the point clouds $P_1$ and $P_2$ will be identical when the term $cd_{P_1 \leftrightarrow P_2}$ is minimized.

We use the Chamfer Distance between $G(P_a)$ and $P_a$ as the autoencoding loss. Beside, the Chamfer Distance from $G(P_r)$ to $P_r$ is utilized as the reconstruction loss, which makes the predicted shape be a part of the input and forces the generator to predict according to the input. We also apply the Chamfer Distance between $R(G(P_a))$ and $P_a$ to optimize the refiner $R$.

In practice, we alternately optimize $G$ with the optimization goal:

$$\min_{\theta_g, \theta_r} \quad \underset{P_i \sim p_r}{\mathbb{E}} \big[ -\alpha_1 \cdot \log \mathcal{P}(P_i; \theta_g, \theta_d) + \alpha_2 \cdot cd_{G(P_i; \theta_g) \to P_i} \big] + \\ \underset{P_i \sim p_a}{\mathbb{E}} [\alpha_3 \cdot cd_{G(P_i; \theta_g) \leftrightarrow P_i} + \alpha_4 \cdot cd_{R(G(P_i; \theta_g); \theta_r) \leftrightarrow P_i}], \tag{8}$$

and $D$ with optimization goal:

$$\min_{\theta_d} \underset{P_i \sim p_r}{\mathbb{E}} \big[ -\alpha_5 \cdot \log \big( 1 - \mathcal{P}(P_i; \theta_g, \theta_d) \big) \big] + \underset{P_i \sim p_a}{\mathbb{E}} \big[ -\alpha_6 \cdot \log \mathcal{P}(P_i; \theta_g, \theta_d) \big], \tag{9}$$

where $\alpha_{1 \sim 6}$ are the weights used for balancing the influences between each term.

## 4 Experiments

### 4.1 Datasets

We first generate the complete point clouds by sampling CAD models of ShapeNet [3] and ModelNet [35], serving as the artificial data. We randomly rotate and scale them to simulate poses of real-scene data. Then we follow a similar process as ScanObjectNN [28] to generate the input point clouds with surrounding points (background points) by cutting out the points in axis-aligned boxes whose origins are the center of objects in ScanNet [7]. The sizes of these axis-aligned boxes are $s$ larger than the axis-aligned bounding boxes of objects, where we set $s$ to $50\%$. We also record labels of points which are 1 for object points and 0 for background points (noises) for evaluation. We select four categories to establish four datasets and randomly split the data into train, valid and test sets. The coordinates of all point clouds are normalized to $[-1, 1]$. Our UGAAN is trained on the train set, and evaluated on test set.

### 4.2 Evaluation Metrics

Generating completion ground truth manually for real-scene data is hard. To overcome this, we utilize Scan2CAD[2], a dataset containing the alignment information from models of ShapeNet to scenes of

Table 1: Comparison (mIOU) of segmentation results by our method (trained with train set of ScanNet) and previous methods on test sets of ScanNet and S3DIS.

| Dataset | Setting | Model | Avg. | Bookcase | Chair | Sofa | Table |
|---|---|---|---|---|---|---|---|
| ScanNet | Full Sup. | PointTransformer [43] | 0.679 | 0.58 | 0.83 | 0.69 | 0.62 |
| | Unsup. | K-Means | 0.201 | 0.25 | 0.23 | 0.18 | 0.16 |
| | | SharinGAN [18] | 0.365 | 0.42 | 0.47 | 0.31 | 0.26 |
| | | Unpaired [39] | 0.422 | 0.47 | 0.58 | 0.34 | 0.30 |
| | | Ours | **0.448** | **0.49** | **0.58** | **0.38** | **0.34** |
| S3DIS | Full Sup. | DGC [32] | 0.722 | 0.34 | 0.96 | 0.69 | 0.9 |
| | | PN++ [24] | 0.735 | 0.44 | 0.92 | 0.75 | 0.83 |
| | Unsup. | K-Means | 0.252 | 0.36 | 0.18 | 0.27 | 0.2 |
| | | AdaCoSeg [45] | 0.31 | 0.34 | 0.24 | 0.38 | 0.28 |
| | | CoSeg [22] | 0.465 | 0.36 | 0.51 | 0.50 | **0.49** |
| | | Unpaired [39] | 0.470 | 0.39 | 0.51 | 0.59 | 0.39 |
| | | Ours | **0.486** | **0.46** | **0.51** | **0.63** | 0.34 |

Table 2: Comparison (CD) of completion results by our method and previous methods on test sets of ScanNet. The results are the smaller the better.

| Model(CD / $F_{score}^{0.1\%}$) | Avg. | Chair | Table | Sofa | Bookcase |
|---|---|---|---|---|---|
| SharinGAN [18] | 0.0322 / 0.091 | 0.0376 / 0.095 | 0.0312 / 0.119 | 0.0224 / 0.111 | 0.0376 / 0.038 |
| Unpaired [39] | 0.0405 / 0.188 | 0.0549 / 0.158 | 0.0339 / 0.274 | 0.0331 / 0.222 | 0.0403 / 0.100 |
| Ours | **0.0274 / 0.217** | **0.0276 / 0.212** | **0.0279 / 0.300** | **0.0211 / 0.233** | **0.0330 / 0.125** |

ScanNet, to replace the incomplete objects with complete models, and use [16, 23, 44] to remove the unseen points caused by occlusion for simulating the real scene data. Then Chamfer Distance(CD) and F-Score@0.1%($F_{score}^{0.1\%}$)[25] is employed to measure the difference between predicted complete point clouds and ground-truth models. For the evaluation of segmentation, we employ IOU, following the previous work [22], to measure the difference between prediction and ground truth.

### 4.3 Implementation details

Our framework is implemented using PyTorch. The point numbers $n_{0\sim2}$ in Section 3 are set to 2048, 512 and 2048, and the weights $\alpha_{1\sim6}$ are set to 1, 100, 5, 100, 0.5 and 0.5. We train four categories separately with the Adam optimizer. The learning rate of the optimizer and the batch size for our network are set to $1.0 \times 10^{-5}$ and 4, separately. We train the model for 240 epochs. Training our UGAAN takes about 20 hours for convergence with a GTX 2080Ti GPU. We will make our dataset and codes public in the future.

### 4.4 Comparison

We compare our UGAAN against existing representative methods on point cloud segmentation including unsupervised methods [22, 39, 45, 18] and supervised methods [43, 32, 24] and on unsupervised point cloud cloud completion [39] both quantitatively and qualitatively. We simply modify [39] by replacing its reconstruction loss with ours and retrain it to fit our datasets, and then utilize our segmentation module to get the segmentation results for evaluation. As shown in Table 1, we compare these methods on ScanNet and S3DIS with four categories, including the chair, table, sofa and bookshelf. Our method is trained on the training set of ScanNet and tested on the test set of these two datasets. The results are 6.16% and 4.51% higher than [39] and [22], showing the superiority of our method. We also test our method trained with ScanNet with more categories on the test set of ScanObjectNN, and the results are shown in the last second row of Table 3. There are only seven common categories in these two datasets. Thus, we can only conduct the comparison on these seven categories. We also train our UGAAN on 15 categories of ScanObjectNN together with ShapeNet and ModelNet to evaluate the generalization ability across multiple categories. The results are shown in the last row of Table 3. The results are 8.73% and 4.46% higher than [22] on "Avg. (part)" and "Avg. (all)". This verifies the generalization ability of our method with different categories. For the quantitative comparison of completion, we build a test set using the method described in Section 4.2

Table 3: Comparison (mIOU) of segmentation results by our and previous methods on test sets of ScanObjectNN. The last two rows show the results of our mothed trained on train set of ScanNet and ScanObjectNN, separately. The "Avg. (part)" indicates the mIOU of the first seven categories (Bookcase ∼ Cabinet) and "Avg. (all)" indicates the mIOU on all the categories.

| Setting | Model | Avg. (part) | Bookcase | Chair | Sofa | Table | Bed | Pillow | Cabinet | Bag | Bin | Box | Desk | Display | Door | Sink | Toilet | Avg. (all) |
|---|---|---|---|---|---|---|---|---|---|---|---|---|---|---|---|---|---|---|
| Full Sup. | DGC [32] | 0.757 | 0.60 | 0.84 | 0.79 | 0.76 | 0.80 | 0.78 | 0.73 | 0.76 | 0.81 | 0.73 | 0.72 | 0.76 | 0.83 | 0.64 | 0.82 | 0.753 |
| | PN++ [24] | 0.764 | 0.62 | 0.84 | 0.81 | 0.77 | 0.81 | 0.75 | 0.75 | 0.75 | 0.83 | 0.79 | 0.77 | 0.79 | 0.83 | 0.68 | 0.85 | 0.775 |
| Unsup. | K-Means | 0.370 | 0.34 | 0.40 | 0.41 | 0.31 | 0.38 | 0.37 | 0.38 | 0.43 | 0.42 | 0.41 | 0.38 | 0.38 | 0.45 | 0.39 | 0.40 | 0.389 |
| | AdaCoSeg [45] | 0.367 | 0.26 | 0.55 | 0.54 | 0.23 | 0.37 | 0.29 | 0.33 | 0.38 | 0.31 | 0.48 | 0.37 | 0.44 | 0.35 | 0.43 | 0.44 | 0.385 |
| | CoSeg [22] | 0.561 | 0.48 | 0.58 | 0.64 | 0.44 | 0.64 | 0.60 | 0.55 | 0.66 | 0.70 | 0.68 | 0.46 | 0.62 | 0.74 | 0.57 | 0.68 | 0.605 |
| | Unpaired [39] | 0.450 | 0.40 | 0.43 | 0.47 | 0.41 | 0.49 | 0.50 | 0.47 | - | - | - | - | - | - | - | - | - |
| | Ours (ScanNet) | **0.610** | **0.53** | **0.73** | **0.64** | **0.53** | **0.66** | **0.61** | **0.56** | - | - | - | - | - | - | - | - | - |
| | Ours (ScanObjectNN) | **0.587** | **0.50** | **0.65** | **0.64** | **0.50** | **0.65** | **0.60** | **0.57** | 0.71 | 0.73 | 0.70 | 0.48 | 0.63 | 0.75 | 0.60 | 0.73 | 0.632 |

Figure 2: Point cloud completion and segmentation results by Unparied [39] and our method. The categories from top to bottom: chair, table, and soda. Red and white in the inputs denote points on the partial objects and outliers, separately. The real inputs do not have the segmentation labels. The left columns shows the real-scenes containing input point clouds.

on the categories containing chair, table, bookcase and sofa. Table 2 shows that our UGAAN can predict more accurate complete point clouds.

Qualitative results on the three categories are shown in Figure 2. The point clouds produced by our UGAAN exhibit more accurate shapes and details, such as the legs and backrest of the chair and the legs of the table shown in Figure 2, leading to more accurate segmentation. And the completion results of ours present more uniform spatial distributions, such as the sofa shown in Figure 2 which contains less noise, which could be attributed to our refiner. Uniform point distribution would facilitate further applications including re-construction, simplification, and so on.

## 4.5 Ablation study

**Ablation study on network architecture.** We conduct experiments on three ablated models to evaluate the necessity of important structures and reconstruction loss of our network. The first model is the basic structure of GAN with a feature extractor and a shape generator. The generator predicts a complete point cloud according to the input point cloud, and the discriminator distinguishes between the prediction and the artificial object. The second model replaces the original generator with the autoencoding generator based on the first model. The third model adds the reconstruction loss introduced in Section 3.3 to the second model. The discriminator of the fourth model makes a distinction between the predictions with real-scene data and the artificial object as input separately. And the discrimination of our full model also takes latent codes as the criterion. As shown in Table 4,

we train and test these ablated models with the ScanNet dataset on the chair category. The evaluation shows the importance of these structures and reconstruction loss. The comparison against the third and fourth models shows that there is a gap between the predictions and artificial point clouds, which explains the reason we apply discriminator to the predictions rather than the artificial point clouds directly.

Table 4: Ablation study of the proposed method.

| Model | IOU |
| --- | --- |
| backbone | 0.122 |
| + autoencoding generator | 0.449 |
| + reconstruction loss | 0.474 |
| + discrimination on predictions | 0.544 |
| + discrimination on latent codes (full) | **0.580** |

Table 5: Ablation study of refiner.

| Dataset | ScanNet | S3DIS | ScanObjectNN |
| --- | --- | --- | --- |
| Ours (no refiner) | **0.447** | 0.478 | 0.596 |
| Ours (full) | 0.444 | **0.486** | **0.610** |

**Refiner.** We also evaluate the importance of refiner both quantitatively and qualitatively. As shown in Figure 3, the refiner can refine the prediction and make the prediction more uniform and reduce noises. The mIOU of our methods without refiner and with refiner on the test sets of the S3DIS and ScanObjNN are shown in Table 5. These methods are trained with the train set of ScanNet. The results show that even though the refiner takes no effect when the method is trained and evaluated on the same dataset, the refiner makes our method more robust when evaluated on new datasets.

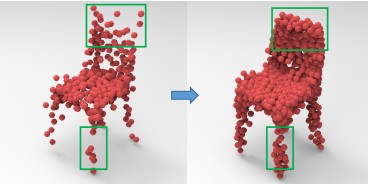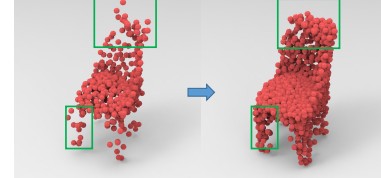

Figure 3: The point clouds before and after refining on chair category.

**The noise ratio $s$.** We also set the noise ratio $s$ to $20\%$, $10\%$, $5\%$ and $0\%$, and generate four more datasets with different noise ratios. Then we repeat the experiments above on the chair category and the IOUs are shown in Figure 4. The lower the noise ratio is, the easier the problem is, making the model perform better. The results also show our generalization performance.

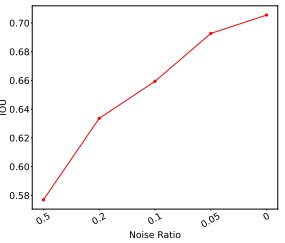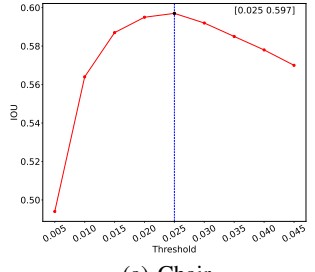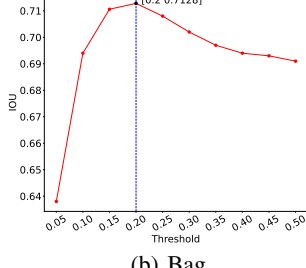

(a) Chair  (b) Bag

Figure 4: The IOU of our method on chair category with different noises ratio.

Figure 5: The IOU of our method with different threshold of intersection on chair and bag categories.

**The threshold $d$ of intersection.** We also evaluate the influence of different threshold $d$ of intersection on the IOU during the evaluation. As Figure 5 shows, we apply different $d$ to the categories chair and bag. The results show that different object has different best $d$, leading to best IOU according to their volume in practice. Since we normalize the coordinates of point clouds to $[-1, 1]$, the point clouds of small objects such as bags contain more details in the same space than big objects such as chairs.

**Extracted features of object points and background points.** We visualize the extracted features of object points and background points from the encoder for analysis. As Figure 6(a) shows, every line indicates the feature of a point. The lighter the pixel is, the larger the value of this channel is. The figure shows that object points have larger values on some channels which can be passed to the latent code by max-pooling operation. Figure 6(b) shows the visualization of the features of artificial points clouds, whose pattern is similar to Figure 6(a), since all the points of artificial data are object points.

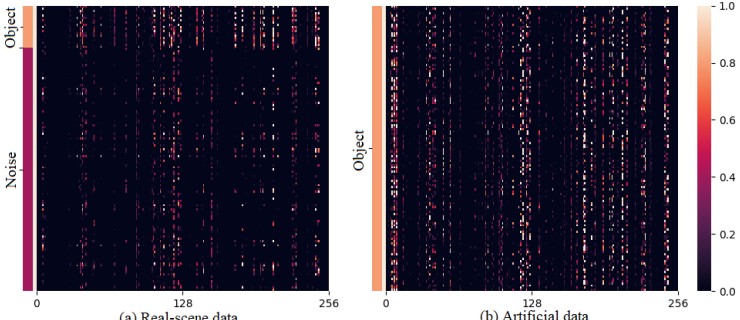

Figure 6: Visualization of the extracted features of real-scene data and artificial data. The bar on the left indicates the type (object or noise) of this line.

**Visualization of latent codes.** We use t-SNE [29] to visualize the latent code distribution of real-scene data and artificial data. As shown in Figure 7 (c), the blue points represent the artificial data and red points indicate the real-scene data, and Figure 7 (e) shows the detailed point clouds of an area. The real-scene data that contains the same shape as the artificial data are grouped together, which shows the robustness of our method with input contaminated by noises. The point clouds having similar yaw angles are grouped together showing that our method could learn reasonable latent codes. This could be attributed to the fact that our discriminator also takes latent codes as the criterion. Visualization of the latent codes extracted from the method whose discriminator only takes predicted point cloud as the criterion is shown in Figure 7 (d). It is obviously observed that these two kinds of data are separated clearly. We also train our method with the dataset that mix up the real-scene data with different noise ratios. As shown in Figure 7 (a) and (b), similar object with different noise ratios are grouped together, which shows the robustness of our method on real-scene data with different noise ratios.

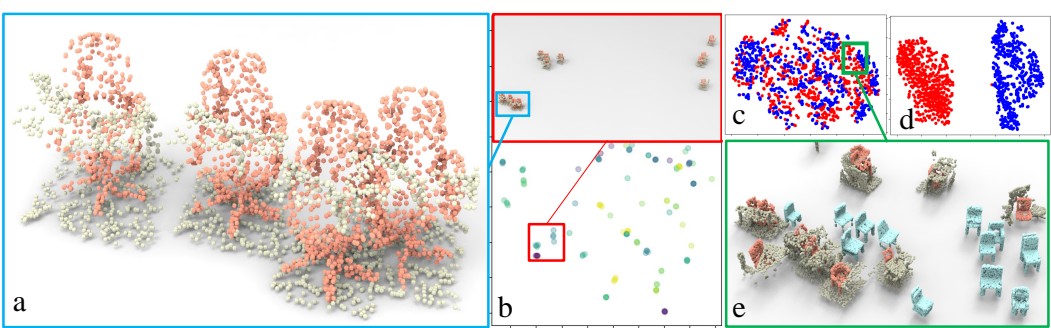

Figure 7: Visualization of t-SNE results with details. The same object with different noise ratio have the same color in Figure (b). Red and blue points are indicate the real-scene data and artificial data, respectively.

## 5   Conclusion, Discussion, and Future Work

We have presented UGAAN, an end-to-end network specifically designed for unsupervised completion and segmentation of point clouds contaminated by noises or containing outliers. Benefiting from our proposed autoencoding generator and latent-code-based discriminator, our method is able to generate complete point cloud accurately and segment the input real-scene point cloud in the meanwhile. Extensive experiments show the superiority and generalization of our method on different categories of different datasets. We also conduct experiments verifying the robustness of our method on noises. Our method needs artificial datasets during the unsupervised training, which limits our method to apply to those categories without artificial correspondences. The encoder and decoder of our generator and the overall structure of GAN can be improved for better prediction and the segmentation module can be replaced with a learning-based module, which are the future work of our method.

## Acknowledgments and Disclosure of Funding

This work was supported in part by the National Natural Science Foundation of China under Grant numbers 62032011, 61972194 and Natural Science Foundation of Jiangsu Province award numbers BK20211147.

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
