# Supplementary Material of Unsupervised Point Cloud Completion and Segmentation by Generative Adversarial Autoencoding Network

In the supplementary material, we first answer several questions mentioned in the checklist. We then show more details about data, training process, network and comparison. Next we conduct more experiments on the generalization of unpaired, multi-category and multi-noise-ratio. Then more qualitative comparisons are given to show the advantages of our UGAAN. Finally, we present the detailed pseudocode of network.

## 1 Answers for Checklist

The datasets that we use contain ScanNet, S3DIS, ScanObjNN, ShapeNet and ModelNet, which are public datasets. We obtained these datasets according to the instructions on their websites. And these datasets are point clouds or meshes data of scenes and objects which contain NO personally identifiable information or offensive content.

## 2 More Details

### 2.1 Data Details

Each entry in our dataset contains an id that identifies the data, an input point cloud with noises, the labels of input point cloud denoting the ground truth of segmentation and an artificial point cloud. The ground truth labels are only used during evaluation. The numbers of models in each category of our generated dataset from ScanNet and coseg's generated dataset from S3DIS and ScanObjectNN are shown in Table 1. The data numbers of corresponding artificial data generated from ShapeNet and ModelNet are also shown in Table 1. As shown in Figure 1(a), the definition of noise ratio is $s = \frac{s_2}{s_1}$. Figure 1(b) and (c) show the difference between our data and the data generated [1] from S3DIS the shape of noise of which are box and sphere separately.

Table 1: Data number of different categories in different dataset.

| Dataset | Bookcase | Chair | Sofa | Table | Bed | Pillow | Cabinet | Bag | Bin | Box | Desk | Display | Door | Sink | Toilet |
|---|---|---|---|---|---|---|---|---|---|---|---|---|---|---|---|
| ScanNet | 1001 | 5425 | 501 | 2581 | | | | | | | | | | | |
| S3DIS | 569 | 1306 | 55 | 455 | | | | | | | | | | | |
| ScanObjectNN | 267 | 395 | 254 | 241 | 135 | 105 | 347 | 77 | 201 | 117 | 149 | 181 | 191 | 118 | 82 |
| ShapeNet | 452 | 4000 | 3173 | 4000 | 233 | 96 | 1571 | 83 | 343 | | | | | | |
| ModelNet | | | | | | | | | | 271 | 286 | 1092 | 129 | 148 | 444 |

### 2.2 Training Details

We use PyTorch to implement our training framework. The generator and the discriminator of our UGAAN are optimized alternately as shown in Algorithm 2. Since different datasets have different data numbers, we train our UGAAN on ScanNet with 240 epochs, and 60 epochs for ScanObjectNN.

**Algorithm 1** The Pseudocode of training framework in PyTorch-like Style

Figure 1: Visualization of our generated data from ScanNet and unpaired generated from S3DIS. Figure (b) is an example generated from ScanNet by us and (c) is an example generated from S3DIS by unpaired. Figure (a) is the bird's eye view of Figure (b) explaining the definition of noise ratio.

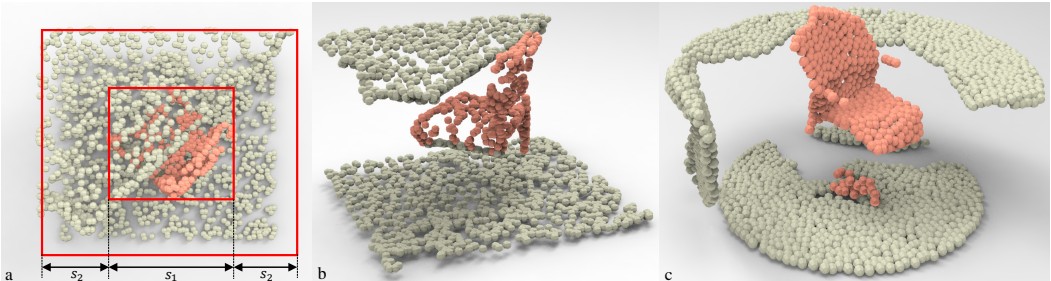

```
def train_one_epoch():
    for data_r, data_a in data:
        pc_r, lc_r, pc_r_r = G(data_r)
        pc_a, lc_a, pc_a_r = G(data_a)
        P_r = D(pc_r, lc_r)
        P_a = D(pc_a, lc_a)
        loss_g = G.loss(P_r, data_a, pc_a, pc_a_r, data_r, pc_r)
        loss_d = D.loss(P_r, P_a)
        opt_G.zero_grad()
        loss_g.backward()
        opt_D.zero_grad()
        loss_d.backward()
        opt_G.step()
        opt_D.step()
```

## 2.3 Network Details

The specific parameters of the weight-shared MLPs [1] used in our CS-Net are clarified in this part. For a MLP, $[c_1, c_2, ..., c_n]$ means that there are $n$ layers in MLP. The input layer has $c_1$ channels, the output layers have $c_n$ channels, and $c_2, ..., c_{n-1}$ denote the channel numbers of hidden layers.

The parameter of the MLP of the decoder in our generator is $[256, 512, 512, 1024, 512 \times 3]$, where $256$ is the channel number of the latent code extracted by the encoder in generator. And the parameter of the MLP in our discriminator is $[1024 + 256, 64, 64, 1]$, where $1024$ is the channel number of the feature extracted by the encoder in discriminator and $256$ indicates the channel number of the latent code.

The pseudo code of our network can be found in Section 6.

## 2.4 Details about Comparison

The original unpaired [3] is designed for incomplete point cloud without any noise, the reconstruction loss of which makes the input point cloud be a part of the prediction. However, our input is incomplete point cloud with severe noise, thus, the original reconstruction loss will make the prediction contain noise. We then compared the original unpaired and its two modified the model with ours as shown in Table 2. The first modified model is removed the reconstruction loss and the second one is replaced the reconstruction loss with ours. The comparison shows the importance of our reconstruction loss. And even so, our UGAAN is still better than Unpaired due to the other carefully designed structures.

---

[1]Weight-shared MLP [2] is widely used in deep learning on point cloud to extract the features. The features of points in a point cloud are generated by the same MLP with the same weight. The MLP is thus called weight-shared MLP.

Table 2: Comparison (mIOU) of results by our method and Unpaired on ScanNet.

| Model | Avg. | Bookcase | Chair | Sofa | Table |
|---|---|---|---|---|---|
| Unpaired (original) | 0.290 | 0.33 | 0.30 | 0.28 | 0.24 |
| Unpaired (no reconstruction loss) | 0.410 | 0.43 | 0.52 | **0.40** | 0.30 |
| Unpaired (our reconstruction loss) | 0.422 | 0.47 | 0.58 | 0.34 | 0.30 |
| Ours | **0.448** | **0.49** | **0.58** | 0.38 | **0.34** |

# 3 More Experiments

## 3.1 The generalization of unpaired

We also evaluate the generalization of unpaired [3]. We train it on our train dataset generated from ScanNet and evaluate on the test set of S3DIS and ScanObjectNN. The comparison are shown in table 3. The results show that our generalization on other datasets is much better than unpaired.

Table 3: The comparison results of unpaired and ours trained with ScanNet on test set of S3DIS and ScanObjectNN.

| Dataset | Model | Avg. | Bookcase | Chair | Sofa | Table | Bed | Pillow | Cabinet |
|---|---|---|---|---|---|---|---|---|---|
| S3DIS | Unpaired | 0.470 | 0.39 | 0.51 | 0.59 | 0.39 | | | |
| | Ours | 0.486 | 0.46 | 0.51 | 0.63 | 0.34 | | | |
| ScanObjectNN | Unpaired | 0.450 | 0.40 | 0.43 | 0.47 | 0.41 | 0.49 | 0.50 | 0.47 |
| | Ours | 0.610 | 0.53 | 0.73 | 0.64 | 0.53 | 0.66 | 0.61 | 0.56 |

## 3.2 Multi-category and Multi-noise-ratio

We also train and evaluate our UGAAN on the dataset that contains all categories(bookcase, chair, sofa and table) with one noise ratio(50%) and all noise ratios (0%, 5%, 10%, 20% and 50%) with one category(chair). The evaluation results on test set are shown in table 4 and table 5. The results also show that our refiner can refine the prediction which makes our UGAAN more robust on the data with different categories and noise ratio.

Table 4: The evaluation results (before and after refining) of our UGAAN trained on the dataset that contains all categories(bookcase, chair, sofa and table) with one noise ratio(50%) are shown in the first two lines. The last line is the evaluation results of our model trained on the dataset that contains a single category.

| Model | Avg. | Bookcase | Chair | Sofa | Table |
|---|---|---|---|---|---|
| Ours(multi-category)(before refining) | 0.408 | 0.44 | 0.48 | 0.38 | 0.33 |
| Ours(multi-category)(after refining) | 0.418 | 0.46 | 0.51 | 0.37 | 0.33 |
| Ours | 0.448 | 0.49 | 0.58 | 0.38 | 0.34 |

## 3.3 The practicality of inputs with random shifts

As noise and errors cannot be avoided in the object detection and instance segmentation process, to better show the practicality of the proposed method, we evaluate our UGAAN on the inputs with random shifts. We add the random shift $x_s \in [-s_{max}, s_{max}]^3$ to all the points of inputs. $s_{max}$ controls the range of the random shift, where we set it to $0, 0.01, 0.02, 0.05, 0.10, 0.20$ for different ranges in the experiments. The mIOUs on the category chair of ScanNet are shown in Table. And Our UGAAN still shows great performs when the $s_{max} < 0.05$.

Table 5: The evaluation results (before and after refining) of our UGAAN trained on the dataset that contains all noise ratio(0%, 5%, 10%, 20% and 50%) with one category(chair) are shown in the first two lines. The last line is the evaluation results of our model trained on the dataset that contains single noise ratio.

| Model | Avg. | 0.5 | 0.2 | 0.1 | 0.05 | 0 |
|---|---|---|---|---|---|---|
| Ours(multi-noise-ratio)(before refining) | 0.556 | 0.55 | 0.57 | 0.56 | 0.55 | 0.55 |
| Ours(multi-noise-ratio)(after refining) | 0.594 | 0.56 | 0.61 | 0.60 | 0.60 | 0.60 |
| Ours | 0.654 | 0.58 | 0.63 | 0.66 | 0.69 | 0.71 |

Table 6: The mIOUs on the category chair of ScanNet where the inputs are randomly shifted with different $s_{max}$.

| $s_{max}$ | 0 | 0.01 | 0.02 | 0.05 | 0.1 | 0.2 |
|---|---|---|---|---|---|---|
| mIOU | 57.79 | 57.58 | 57.33 | 54.04 | 44.41 | 23.94 |

# 4 Error bars of experiments

The error bars of our training experiments are shown in Figure 2(a) and 2(b), which shows that the performance of our UGAAN is stable and easy to reproduce.

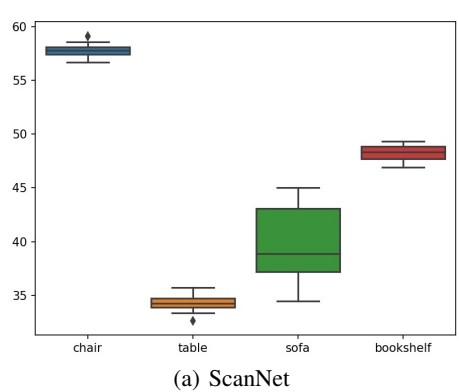
(a) ScanNet

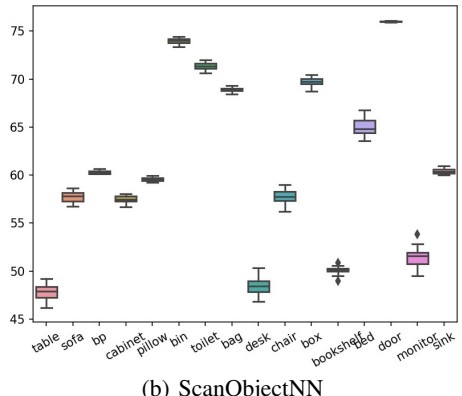
(b) ScanObjectNN

Figure 2: The box image of evaluation results of our model trained on ScanNet and ScanObjectNN, where diamonds indicate the outliers.

# 5 More Qualitative Comparisons

More qualitative comparisons of our UGAAN that are trained on the train set of ScanNet and evaluated on ScanNet, S3DIS and ScanObjectNN are shown in Figures 3, 4,5 and 6.

# 6 Pseudo code

The detailed pseudocodes of our UGAAN written in PyTorch-like style are shown in Algorithms.

**Algorithm 2** The Pseudocode of UGAAN in PyTorch-like Style

```
class Generator:
    def __init__(self):
        self.encoder = PN2()
        self.decoder = MLP()
        self.encoder_r = PCNEncoder()
        self.decoder_r = PCNDecoder()
```

```
        self.loss = G_loss()
    def forward(self, p_input):
        # p_input: Bx3x2048
        lc = self.encoder(p_input)  # Bx256x1
        pc = self.decoder(lc)       # Bx3x512
        gf = self.encoder_r(pc)     # Bx1024x1
        pc_r = self.decoder_r(gf, pc)   # Bx3x2048
        return pc, lc, pc_r

class Discriminator:
    def __init__(self):
        self.encoder = PCNEncoder()
        self.decoder = MLP()
        self.loss = D_loss()
    def forward(self, pc, lc):
        # pc:   Bx3x512
        # lc:   Bx256x1
        f = self.encoder(pc)     # Bx1024x1
        f = self.cat([f, lc], 1)     # Bx(1024+256)x1
        P = self.decoder(f)      # Bx1x1
        return P

class G_loss:
    def __init__(self):
        self.cd = ChamferDistance()
    def CD(self, pc1, pc2):
        return self.cd(pc1, pc2) + self.cd(pc2, pc1)
    def forward(self, P_r, data_a, pc_a, pc_a_r, data_r, pc_r):
        return -log(P_r) + 1e2*CD(data_a, pc_a) + 1e2*self.CD(data_a,
    pc_a_r) + 5*self.cd(pc_r, data_r)

class D_loss:
    def forward(self, P_r, P_a):
        return -0.5*(log(1-P_r) + log(P_a))
```

Figure 3: Visualization of segmentation and completion results of chairs and sofas on ScanNet and S3DIS.

| Input | Seg. | Com. | Input | Seg. | Com. |

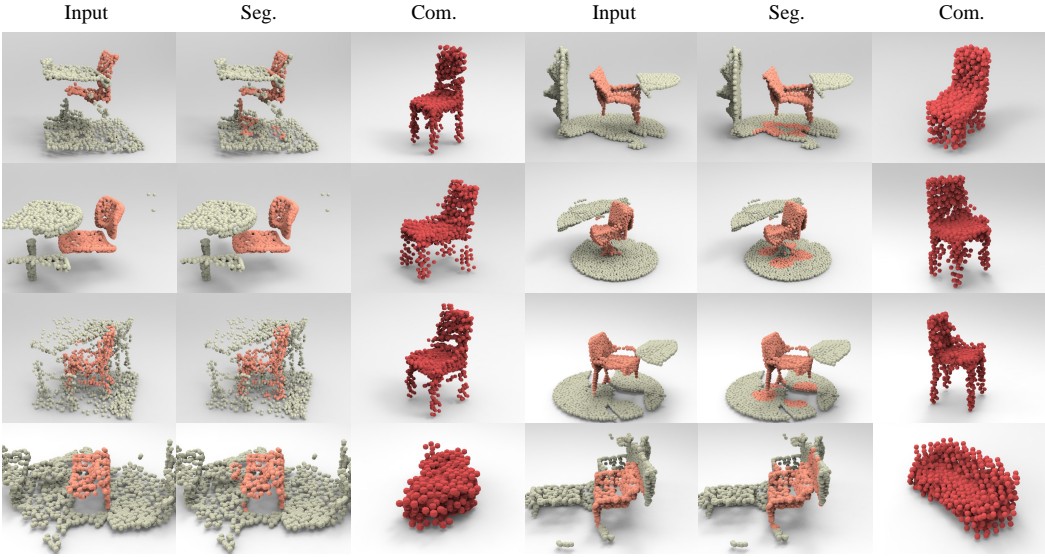

Figure 4: Visualization of segmentation and completion results of tables on ScanNet and S3DIS.

Input  Seg.  Com.  Input  Seg.  Com.

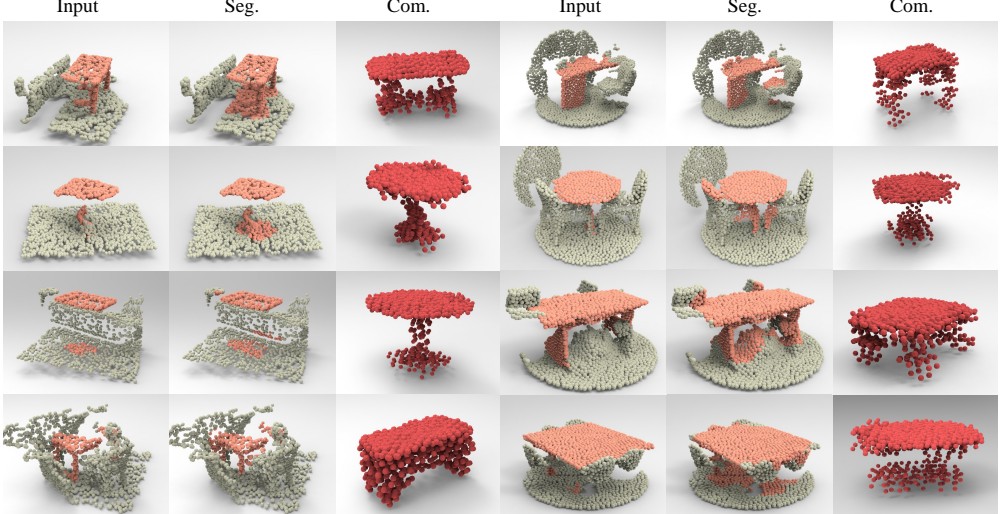

Figure 5: Visualization of segmentation and completion results of bookcases on ScanNet and S3DIS.

Input  Seg.  Com.  Input  Seg.  Com.

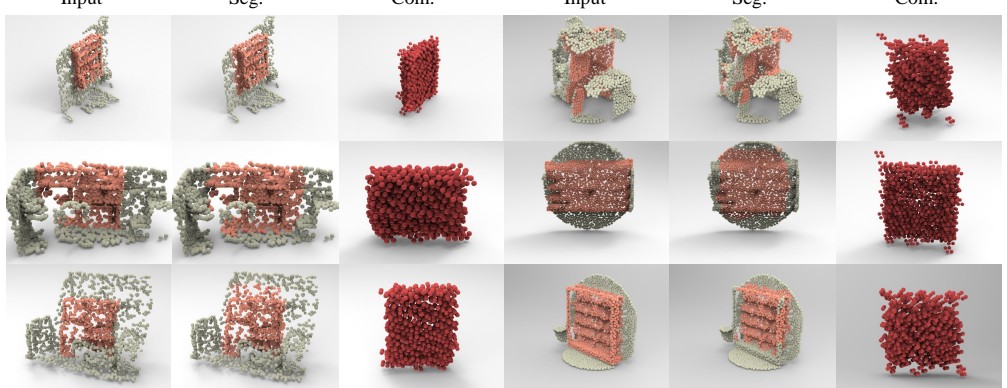

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

Figure 6: Visualization of segmentation and completion results on ScanObjectNN. From top to bottom, from left to right, the models are chair, sofa, bed, table, cabinet, bookcase, pillow and bookcase.

| Input | Seg. | Com. | Input | Seg. | Com. |
|-------|------|------|-------|------|------|

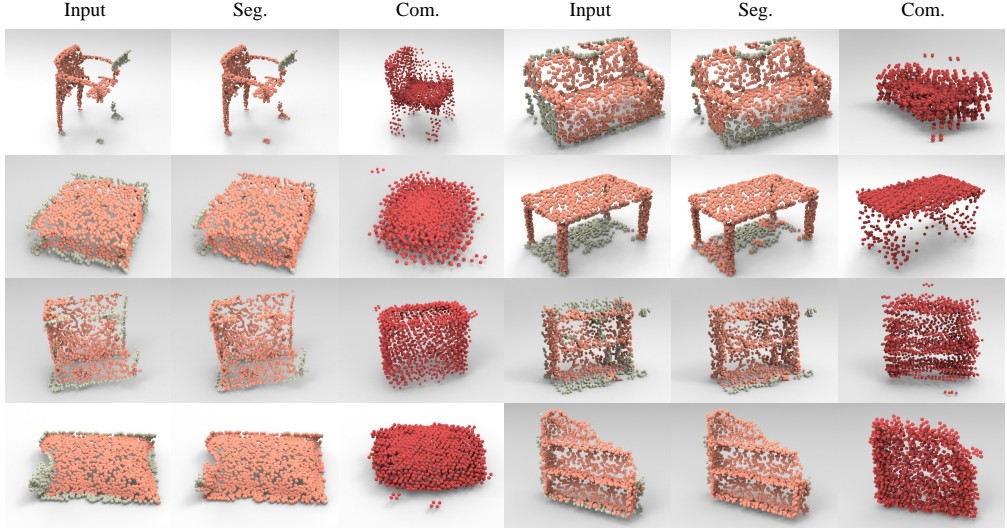