# OpenReview forum: "Unsupervised Point Cloud Completion and Segmentation by Generative Adversarial Autoencoding Network"
_NeurIPS.cc/2022/Conference — NeurIPS 2022 Accept_

### Official Review · Reviewer_bh96 · 2022-07-08

**Rating:** 6
**Confidence:** 4
**Soundness:** 3 good
**Presentation:** 3 good
**Contribution:** 3 good

**Summary:**

This work proposes an unsupervised point cloud completion and segmentation method in the case that the input point cloud contains background clutters or noises. The core idea of the method is to train an autoencoder and a GAN at the same time between the corrupted real point cloud data to complete and segment, and the complete CAD model data. The authors conducted experiments over the ShapeNet, ScanNet, S3DIS, and ScanObjectNN data sets, and compared the proposed method to several previous works as baselines where superior segmentation performance is demonstrated. Extensive ablation studies are also presented to validate several network design details.

**Questions:**

I only have two questions below, but they are major concerns. I will consider raising up my score if the two questions are well addressed.
- Can you show the setting where instance segmentators are used or justify why not do this experiment?
- Can you add quantitative evaluation and comparisons regarding the shape completion performance?

**Limitations:**

The authors have discussed the limitations and I don't have further concerns.

**Strengths And Weaknesses:**

Strengths:
- the proposed method contains several non-trivial technical designs, such as training the discriminators over the AE generated real and fake data, regularizing the GAN training with some reconstruction losses, feeding the latent codes to the discriminators as well, how to perform the segmentation, etc. The proposed method is valid, sound and innovative.
- the authors conducted quite extensive experiments by using three popular scene datasets (ScanNet, ScanObjectNN, and S3DIS) and compared to several previous works as baselines, showing superior segmentation performance. The experiments are quite solid and convincing.

Weaknesses:
- the reviewer wants to question the assumption that only object bounding box information can be used to crop out the object with background clutters and noises. There are so many powerful instance segmentation methods, say Mask-RCNN and many 3D variants, which can crop out the foreground objects much better and cleaner than using only the bounding boxes as the cropper. While the reviewer agrees that these methods have failure cases, I believe that it's important to show experiments in these settings where the instance segmentation masks are used to crop out the objects and validate if the proposed method still demonstrates better results in these settings where the noises are much less and the data is much easier, maybe for competitive baseline methods such as [6]. I think many point cloud completion methods may assume using such instance segmentators to crop out the input object point clouds and also demonstrated experiments where certain noises are added to the input point clouds. Line 30-32 tries to argue that instance segmentation is hard, but please show it. I have big concerns regarding the input data assumption used in this paper.
- Another big concern of mine is that though the paper claims in the title and the story that the method is doing well for completion there is actually no quantitative evaluation and comparison for the completion performance. The paper mostly evaluates regarding the segmentation performance. While the reviewer understands that it's impossible to get the ground-truth data for real-scene data sets, I still think there should be quantitative comparisons. At least the authors can show user studies results over a large scale of test data points. I think it's also helpful to use some synthetic data just to show some quantitative comparisons to baseline methods.
- Table 1 and 2 does not include quantitative comparisons to [6], which is one of the most competitive method. The reviewer finds the comparisons in the supplementary material, but I think it's important to move them in the main paper.

---

> ### Author Response · Authors · 2022-08-01
> **For Reviewer bh96**
>
> We thank Reviewer bh96 for valuable and constructive comments.
>
> **(1) The setting using instance segmentators:** We also agree that the setting using instance segmentators which contains much fewer noises is much easier for baseline methods. So we use the inputs that contain no noises to simulate this setting in the extreme situation. We conduct new experiments to compare our UGAAN and Unpaired[6] trained on the original data and retrained on data with no noises, and the results are shown in table below. The comparison shows that our method still demonstrates better results in the setting using instance segmentators. Meanwhile, the results of our method trained with the original data also show our robustness on noises.
>
> ***
> |                          | AVG    | Chair  | Table  | Sofa   | Bookcase  |
> | ------------------------:|:------:|:------:|:------:|:------:|:---------:|
> | Unpaired(original data)  | 0.0525 | 0.0622 | 0.0466 | 0.0417 | 0.0595    |
> | Unpaired(no noises data) | 0.0365 | 0.0503 | 0.0257 | 0.0292 | 0.0411    |
> | Ours(original data)      | 0.0250 | 0.0200 | 0.0242 | 0.0276 | 0.0283    |
> | Ours(no noises data)     | **0.0165** | **0.0175** | **0.0160** | **0.0142** | **0.0185**    |
> ***
>
> **(2) Quantitative evaluation and comparison of completion:** As suggested, we conduct new experiments to quantitatively evaluate our method and compare our UGAAN with the Unpaired [6] on the completion results. Since getting the ground-truth for real-scene data is impossible, as Reviewer bh96 pointed out, we utilize Scan2CAD[r1], a dataset containing the alignment information from models of ShapeNet to ScanNet scenes, to replace the incomplete objects with complete models, and use [r2, r3, r4] to remove the unseen points caused by occlusion for simulating real scenarios. We can easily generate the complete ground-truths and simulated inputs in this way. We then build a test set on the categories containing chair, table, bookcase and sofa for evaluation and comparison, where methods are trained using the original dataset. The comparisons using Chamfer Distance and F1-Score\@0.1%[r5] for evaluation are shown in the tables below. The results of Chamfer Distance are the smaller the better and the results of F-Score\@0.1% are the higher the better. The quantitative comparison shows the superiority of our UGAAN. We also put the comparisons into our main paper. Please see Table 2 on Page 6, marked in red for indicating the revision, of our submitted rebuttal version..
> ***
> | CD          | AVG    | Chair  | Table  | Sofa   | Bookcase  |
> |:-----------:|:------:|:------:|:------:|:------:|:---------:|
> | Unpaired[6] | 0.0405 | 0.0549 | 0.0339 | 0.0331 | 0.0403    |
> | Ours        | **0.0274** | **0.0276** | **0.0279** | **0.0211** | **0.0330**    |
> ***
> | F-Score\@0.1% | AVG   | Chair | Table | Sofa  | Bookcase  |
> |:------------:|:-----:|:-----:|:-----:|:-----:|:---------:|
> | Unpaired[6]  | 0.188 | 0.158 | 0.274 | 0.222 | 0.100     |
> | Ours         | **0.217** | **0.212** | **0.300** | **0.233** | **0.125**     |
> ***
>
> **(3) Move the comparisons between the proposed method and the Unpaired into the main paper:** We integrate Tables 2 and 3 of our supplementary material into Tables 1 and 3 of our main paper. Please see Pages 6 and 7, marked in red for indicating the revision, of our submitted rebuttal version.
>
> [6] Niloy J. Mitra Xuelin Chen, Baoquan Chen.  Unpaired point cloud completion on real scans using adversarial training. In2020 International Conference on Learning Representations 2020 (ICLR), 2020
>
> [r1] Armen Avetisyan, Manuel Dahnert, Angela Dai, Manolis Savva, Angel X. Chang, and Matthias Niessner. Scan2cad: Learning cad model alignment in rgb-d scans. In The IEEE Conference on Computer Vision and Pattern Recognition (CVPR), June 2019. 2, 6. https://github.com/skanti/Scan2CAD
>
> [r2] Ayellet Tal Katz Sagi and Ronen Basri. Direct visibility of point sets. ACM Transactions on Graphics (TOG), page 24, July 2007. 6
>
> [r3] Ravish Mehra, Pushkar Tripathi, Alla Sheffer, and Niloy J. Mitra. Visibility of noisy point cloud data. Computers and Graphics, In Press, Accepted Manuscript:–, 2010. 6
>
> [r4] Qian-Yi Zhou, Jaesik Park, and Vladlen Koltun. Open3D: A modern library for 3D data processing. arXiv:1801.09847, 2018. 6
>
> [r5] Tatarchenko, M., Richter, S.R., Ranftl, R., Li, Z., Koltun, V., Brox, T.: What dosingle-view 3d reconstruction networks learn In: Proceedings of the IEEE/CVFConference on Computer Vision and Pattern Recognition. pp. 3405–3414 (2019)

---

> > ### Comment · Reviewer_bh96 · 2022-08-10
> > **thanks and changed to 6**
> >
> > thanks for the author rebuttal which addressed my two major concerns. I'd like to change my score to 6.

---

> > > ### Author Response · Authors · 2022-08-10
> > > **Thank you for your decision!**
> > >
> > > Thanks for the support to our paper!

---

### Official Review · Reviewer_ujzp · 2022-07-11

**Rating:** 6
**Confidence:** 4
**Soundness:** 3 good
**Presentation:** 3 good
**Contribution:** 3 good

**Summary:**

This paper proposed a GAN to generate a complete point cloud from real scanned point clouds with the help of artificial CAD models. The completed point clouds are also used to segment objects from the background. Experimental results show the proposed method performs well in segmentation and demonstrates the proposed method can learn cross-domain information.

**Questions:**

Please address the above weaknesses. Besides, I think comparisons between the proposed method and the Unpaired are essential. I would suggest the authors move them into the paper.

**Limitations:**

The authors have not addressed the limitations or potential negative societal impact. My suggestions are listed in the weaknesses parts.

**Strengths And Weaknesses:**

Strengths:
1.	Achieves some kinds of unsupervised foreground-background point clouds segmentation.
2.	The problem of noisy point cloud completion is interesting and valuable.
3.	Method descriptions are clear and easy to understand.
4.	Experiments about segmentation and latent codes distribution are sufficient and informative
5.	The overall architecture is reasonable and sound.

Weaknesses:
1.	The input of the mentioned segmentation requires the object located at the center. I think this requirement is too hard to fulfill in the real world, and the proposed method didn’t consider this.
2.	Although the paper is focusing on point clouds completion, statistics and comparison of completion performance are missing in the paper.

---

> ### Author Response · Authors · 2022-08-01
> **For Reviewer ujzp**
>
> We thank Reviewer ujzp for valuable and constructive comments.
>
> **(1) The location of input objects:** As we mentioned in the Introduction of the paper, the inputs of our UGAAN can be the outputs of object detection or instance segmentation, which are easy to apply in the real world. The target objects from these outputs are actually in the center of the whole point cloud. And please note that such an experimental setting is also used by CoSeg [21] which also requires the target objects need to be in the center of input point clouds.
>
> **(2) Quantitative evaluation and comparison of completion:** As suggested, we conduct new experiments to quantitatively evaluate our method and compare our UGAAN with the Unpaired [6] on the completion results. Since getting the ground-truth for real-scene data is impossible, as Reviewer bh96 pointed out, we utilize Scan2CAD[r1], a dataset containing the alignment information from models of ShapeNet to ScanNet scenes, to replace the incomplete objects with complete models, and use [r2, r3, r4] to remove the unseen points caused by occlusion for simulating real scenarios. We can easily generate the complete ground-truths and simulated inputs in this way. We then build a test set on the categories containing chair, table, bookcase and sofa for evaluation and comparison, where methods are trained using the original dataset. The comparisons using Chamfer Distance and F1-Score\@0.1%[r5] for evaluation are shown in the tables below. The results of Chamfer Distance are the smaller the better and the results of F-Score\@0.1% are the higher the better. The quantitative comparison shows the superiority of our UGAAN. We also put the comparisons into our main paper. Please see Table 2 on Page 6, marked in red for indicating the revision, of our submitted rebuttal version..
> ***
> | CD          | AVG    | Chair  | Table  | Sofa   | Bookcase  |
> |:-----------:|:------:|:------:|:------:|:------:|:---------:|
> | Unpaired[6] | 0.0405 | 0.0549 | 0.0339 | 0.0331 | 0.0403    |
> | Ours        | **0.0274** | **0.0276** | **0.0279** | **0.0211** | **0.0330**    |
> ***
> | F-Score\@0.1% | AVG   | Chair | Table | Sofa  | Bookcase  |
> |:------------:|:-----:|:-----:|:-----:|:-----:|:---------:|
> | Unpaired[6]  | 0.188 | 0.158 | 0.274 | 0.222 | 0.100     |
> | Ours         | **0.217** | **0.212** | **0.300** | **0.233** | **0.125**     |
> ***
>
> **(3) Move the comparisons between the proposed method and the Unpaired into the main paper:** We integrate Tables 2 and 3 of our supplementary material into Tables 1 and 3 of our main paper. Please see Pages 6 and 7, marked in red for indicating the revision, of our submitted rebuttal version.
>
>
> [21] Cheng-Kun Yang; Yung-Yu Chuang; Yen-Yu Lin. Unsupervised point cloud object co-segmentation by co-contrastive learning and mutual attention sampling. In 2021 IEEE/CVF International Conference on Computer Vision (ICCV), 2021
>
> [6] Niloy J. Mitra Xuelin Chen, Baoquan Chen.  Unpaired point cloud completion on real scans using adversarial training. In2020 International Conference on Learning Representations 2020 (ICLR), 2020
>
> [r1] Armen Avetisyan, Manuel Dahnert, Angela Dai, Manolis Savva, Angel X. Chang, and Matthias Niessner. Scan2cad: Learning cad model alignment in rgb-d scans. In The IEEE Conference on Computer Vision and Pattern Recognition (CVPR), June 2019. 2, 6. https://github.com/skanti/Scan2CAD
>
> [r2] Ayellet Tal Katz Sagi and Ronen Basri. Direct visibility of point sets. ACM Transactions on Graphics (TOG), page 24, July 2007. 6
>
> [r3] Ravish Mehra, Pushkar Tripathi, Alla Sheffer, and Niloy J. Mitra. Visibility of noisy point cloud data. Computers and Graphics, In Press, Accepted Manuscript:–, 2010. 6
>
> [r4] Qian-Yi Zhou, Jaesik Park, and Vladlen Koltun. Open3D: A modern library for 3D data processing. arXiv:1801.09847, 2018. 6
>
> [r5] Tatarchenko, M., Richter, S.R., Ranftl, R., Li, Z., Koltun, V., Brox, T.: What dosingle-view 3d reconstruction networks learn In: Proceedings of the IEEE/CVFConference on Computer Vision and Pattern Recognition. pp. 3405–3414 (2019)

---

> > ### Comment · Reviewer_ujzp · 2022-08-08
> > **Final Rating**
> >
> > The authors have solved most of my concerns. As noise and errors cannot be avoided in the object detection and instance segmentation process, to better show the practicality of the proposed method, I would strongly suggest authors add some experiments on the relationship between mIOU and the spatial offsets of instances.
> > As all my major concerns are resolved, I will keep my rating as 6: Weak accept.

---

> > > ### Author Response · Authors · 2022-08-09
> > > **Thank you for your decision!**
> > >
> > > Thanks for the support to our paper! As suggested, we will add some experiments on the relationship between mIOU and the spatial offsets of instances to the final version, to better show the practicality of the proposed method.

---

### Official Review · Reviewer_hDJC · 2022-07-18

**Rating:** 5
**Confidence:** 2
**Soundness:** 3 good
**Presentation:** 3 good
**Contribution:** 2 fair

**Summary:**

They propose UGAAN unsupervised end-to-end network for completing the partial point cloud with noises from real scenes. The motivation is that supervised point cloud completion and segmentation require the paired clean and complete point clouds, together with object-level point labels of real scenes, which is hard to obtain in real settings. They also propose using an autoencoding generator to learn the basic shapes from the artificial data, and  to help decrease the domain gap between real-scene data and make the optimization more stable. They show results on ScanNet with ShapeNet as the artificial dataset and evaluate it on the other two real-scene datasets, including S3DIS and ScanObjectNN.


**Questions:**

I think the method makes sense and has good results, but are few baselines which should be added such as SharinGAN which will make the paper much stronger.

**Limitations:**

Yes,  authors adequately addressed the limitations and potential negative societal impact of their work.

**Strengths And Weaknesses:**

Strengths:
1) The overall intuition behind the problem makes a lot of sense, collecting real world data for point cloud completion on large scale is a very challenging task and UGAAN can help in reducing the amount of required data for point cloud completion.
2) The idea of an autoencoding generator is novel; their discriminator accepts the prediction of artificial data rather than the artificial data itself as real data, which helps in reducing the domain gap.
3) They show decent results on ScanNet and transfer well to two real-scene datasets.  They also show ablation study for their method and for the refiner as well.

Weakness:
1) The idea of decreasing the domain gap between synthetic and real datasets is a separate field of research. For example, SharinGAN[1] also tries to decrease the domain gap. I think there should be comparison to more baselines, and at the very least there should be comparison to SharinGAN and how UGAAN performs wrt to SharinGAN[1].
2) Also, the GAN used currently is the Vanilla GAN, there have been a lot of GAN’s that have come around which have much better performance than the Vanilla GAN. Instead of using Vanila GAN , maybe something like StyleGAN should be used which would probably give much better results.

[1] SharinGAN: Combining Synthetic and Real Data for Unsupervised Geometry
Estimation. https://openaccess.thecvf.com/content_CVPR_2020/papers/PNVR_SharinGAN_Combining_Synthetic_and_Real_Data_for_Unsupervised_Geometry_Estimation_CVPR_2020_paper.pdf

---

> ### Author Response · Authors · 2022-08-01
> **For Reviewer hDJC**
>
>
> We thank Reviewer hDJC for valuable and constructive comments.
>
> **(1) The comparison between our UGAAN and SharinGAN:** As suggested, we conduct new experiments to compare with another baseline, SharinGAN[r6], a work focusing on domain adaptation. The comparisons of completion and segmentation are shown in the tables below. We employ Chamfer Distance (CD) and F-Score\@0.1%[r5] to evaluate the completion results, and IOU to evaluate the segmentation results. The results of Chamfer Distance are the smaller the better and the results of F-Score@0.1% and IOU are the higher the better. ShairnGAN is designed for domain-gap of images which is an easier problem compared with point clouds. Our UGAAN is specially designed for point clouds, and thus performs better than SharinGAN as shown in comparisons, indicating that our UGANN shows effectiveness and superiority in handling the domain-gap between real-scene data and artificial data. We also put the comparisons into our main paper. Please see Tables 1 and 2 on Page 6, marked in red for indicating the revision, of our submitted rebuttal version.
> ***
> | COM.(CD)      | AVG.   | Chair  | Table  | Sofa   | Bookcase  |
> |:---------:|:------:|:------:|:------:|:------:|:---------:|
> | SharinGAN | 0.0322 | 0.0376 | 0.0312 | 0.0224 | 0.0376    |
> | Ours      | **0.0274** | **0.0276** | **0.0279** | **0.0211** | **0.0330**    |
> ***
> | COM.( F-Score\@0.1%) | AVG   | Chair | Table | Sofa  | Bookcase  |
> |:-------------------:|:-----:|:-----:|:-----:|:-----:|:---------:|
> | SharinGAN           | 0.091 | 0.095 | 0.119 | 0.111 | 0.038     |
> | Ours                | **0.217** | **0.212** | **0.300** | **0.233** | **0.125**     |
> ***
> | SEG.(IOU)       | AVG.    | Chair | Table | Sofa | Bookcase  |
> |:---------:|:------:|:-----:|:-----:|:----:|:---------:|
> | SharinGAN | 0.36  | 0.47  | 0.26  | 0.31 | 0.42      |
> | Ours      | **0.46** | **0.58**  | **0.34**  | **0.38** | **0.49**      |
> ***
> Please note that generating complete ground truth manually for real-scene data to evaluate the completion results is hard. Reviewer bh96 pointed out we cloud use synthetic data for evaluation. Thus, we utilize Scan2CAD[r1], a dataset containing the alignment information from models of ShapeNet to scenes of ScanNet, to replace the incomplete objects with complete models, and use [r2, r3, r4] to remove the unseen points caused by occlusion for simulating the real scene data. We can easily generate the complete ground-truths and simulated inputs in this way.
>
> **(2) The improvement of novel GAN's structure:** Our UGANN outperforms state-of-the-arts which aims to realize unsupervised point cloud completion and reduce the gap between real-scene data and artificial data. Our UGAAN benefits from serval important components: the autoencoding GAN with refiner for learning the basic shapes and refining details of prediction, the discrimination on both predictions and latent codes for learning better latent codes of partial objects with noises, the segmentation module using complete prediction for segmentation and the carefully designed losses for keeping information of inputs. The contributions lie in that we prosed UGAAN consisting of above components and shows generalization and robustness on different datasets without pre-training, whose evaluation results are 6.16% and 15.4% higher than recent state-of-the-arts on segmentation and completion. Theoretically, our framework is compatible with different structures of GANs. In our current implementation, we use the Vanila Gan structure since it is not our contribution. Though not complex, our current framework with the Vanila GAN shows effectiveness and outperforms state-of-the-arts. Thus, in our current implementation, we use the Vanila Gan structure since it is not our contribution. We agree with the Reviewer that recent novel structures of GAN would probably help improve the performance of our network, and actually it is a great idea. As the Reviewer said, we have tried to add the Latent Code Mapping Network of StyleGAN to our UGAAN. Unfortunately, finding the best way to fuse this structure into our network is not trivial. But combining the novel structure of recent GANs and getting better performance will be our future work.
>
> [r1] Armen Avetisyan: Scan2cad: Learning cad model alignment in rgb-d scans. CVPR, June 2019. 2, 6 https://github.com/skanti/Scan2CAD
>
> [r2] Ayellet Tal: Direct visibility of point sets. ACM TOG, page 24, July 2007. 6
>
> [r3] Ravish Mehra: Visibility of noisy point cloud data. Computers and Graphics, 2010. 6
>
> [r4] Qian-Yi Zhou: Open3D: A modern library for 3D data processing. arXiv:1801.09847, 2018. 6
>
> [r5] Tatarchenko: What dosingle-view 3d reconstruction networks learn In: Proceedings of the CVPR. pp. 3405–3414 (2019)
>
> [r6] P. N. V. R. Koutilya. Sharingan: Combining synthetic and real data for unsupervised geometry estimation. CVPR, pages 13971–13980, 2020

---

> > ### Comment · Reviewer_hDJC · 2022-08-10
> > **Final Rating**
> >
> > Thanks for the experimental comparison with SharinGAN. I'm raising my score to 5. However, I still feel there should be more experimental comparison to other baselines especially methods decreasing domain gaps. I'm also not really convinced why about why still vanilla GANs are being used instead of something like StyleGAN.

---

> > > ### Author Response · Authors · 2022-08-10
> > > **Thank you for your decision!**
> > >
> > > Thanks! We will try to find out more solutions successfully used to solve other problems involving doman gap and compare with them if possible. Further, we will try to experiment with more advanced GANs as time permits. The new experiments will be included in the final version.

---

### Meta-Review · Area_Chair_Mifp · 2022-08-29

**Recommendation:** Accept
**Confidence:** Certain

**Metareview:**

This paper proposes to leverage generative models (GAN) to perform cross-domain data generation for point cloud completion and segmentation applications. All three reviewers voted to accept the paper, citing novelty, good writing and convincing experimental results. Authors are requested to take into account reviewer feedback and incorporate into their final version. Congratulations!

**Award:**

No

---

### Decision · Program_Chairs · 2022-09-14

Accept